# Three-Party Stochastic Evolutionary Game Analysis of Supply Chain Finance Based on Blockchain Technology

**Qingfeng Zhu** [1]🔘**, Rui Zong** [2,*]🔘 **and Mengqi Xu** [2]

1 Shandong Key Laboratory of Blockchain Finance, School of Statistics and Mathematics, Shandong University of Finance and Economics, Jinan 250014, China
2 School of Statistics and Mathematics, Shandong University of Finance and Economics, Jinan 250014, China
* Correspondence: zongrui298@163.com

**Abstract:** In the process of accounts receivable financing under supply chain finance, the phenomenon of accounts receivable forgery and default have caused great pressure on the supervision of financial institutions. We consider the integration of blockchain technology with a supply chain finance platform around the fraudulent default phenomenon in supply chain finance receivables financing and construct a three-party stochastic evolutionary game model among financial institutions, core enterprises, and Micro, Small, and Medium Enterprises (MSMEs). Firstly, we use *Itô*'s stochastic differential equation theory to analyze the conditions for the stability of the behavior of game subjects. Secondly, we use numerical simulations to quantitatively analyze the impact of the regulatory strength of financial institutions, the information sharing of the blockchain platform, and the change of incentive parameters on the strategy choice of game subjects. Through the above analysis, we conclude that the information-sharing incentive coefficient promotes financial institutions to choose to connect to the blockchain platform, and the information-sharing risk coefficient and the regulatory intensity have the opposite effect on the blockchain platform construction. Meanwhile, the allocation of incentive shares has a significant influence on the core enterprises. Finally, we give priorities and directions for adjusting the relevant parameters to provide recommendations for financial institutions to regulate the financing process more effectively.

**Keywords:** blockchain; supply chain finance; stochastic evolutionary game; accounts receivable financing

## 1. Introduction

As an important part of China's market economy, Micro, Small, and Medium Enterprises (MSMEs) play an important role in the process of China's economic development. However, due to their qualifications, credit credentials, and proof of property, MSMEs often have difficulties in obtaining low-cost loans from formal channels due to their low credit assessment ratings in commercial banks and other financial institutions. As the upstream of the supply chain, MSMEs are often short of funds for their production activities, and their inability to obtain financing seriously affects their liquidity, which can easily lead to a break in their capital chain and thus affect the operation of the entire supply chain [1]. The emergence of supply chain finance has brought a solution to the problem of financing difficulties for MSMEs in the supply chain. Through a secured credit model based on real trade, the core supply chain enterprises guarantee their partners, and the MSMEs thus obtain loans from financial institutions or fintech companies [2]. However, even with a guarantee from the core enterprise, new problems, such as slow information, high costs, and the risk of fraudulent lending, arise. Due to the multi-level structure of the supply chain, there are information barriers and information silos for enterprises at the tail end, data are often poorly transmitted, the intermediate offline process is complex

and inefficient, the cost of seeking evidence is high, and MSMEs and core enterprises may falsify transaction information for fraudulent lending. For example, from September to November 2016, Genertec Dalian Machine Tool falsified its transaction information with BYD by falsifying accounts receivable, forging contracts and official seals, and so on, to obtain CNY 600 million from Zhongjiang Trust.

The features of blockchain can be a good solution to several problems that arise in supply chain finance and in theory can reduce the risks currently faced by supply chain finance [3]. Blockchain technology is a distributed ledger technology established in a non-secure environment, which allows all participants to become a node in the chain, and each node can keep accounts on it and pack all the accounting information into a data block by cryptographic algorithm records in the chronological order of accounting, using cryptographic technology and asymmetric encryption to ensure the security of the data [4]. To verify the validity of the information, all the nodes in the chain will confirm the authenticity of the data block information together. The blockchain platform integrates the asset cooperation, trade transactions, and closely related proof and evidence of each enterprise in the supply chain into one chain, breaking through the existing three-party credit model, achieving a decentralized and intermediary purpose, improving the efficiency of capital operation, reducing transaction costs, and improving the convenience of financing. For example, ICBC has built the ICBC e-Credit network financing and financial services platform to mitigate risk through the automatic execution of smart contracts and a new and reliable supply chain credit system. Tencent Cloud integrated blockchain technology with warehouse receipt pledge financing scenarios, solving the problems of forgery, multiple pledges, and false collateral in the traditional warehouse receipt pledge financing process, realizing data-based management of the pledge transaction process.

In the above context, the application of blockchain technology in the receivables financing process of supply chain finance is quite important. However, the development and application of blockchain technology are still in the fumbling stage in domestic industry due to the cost of information and the inefficiency of information exchange. In reality, several problems concerning blockchain technology and supply chain finance need to be solved. In this article, we explore the following issues:

(1) What factors influence whether financial institutions require supply chain companies to interface with blockchain technology platforms?
(2) What factors influence the repayment of core enterprises and MSMEs?
(3) How should financial institutions work with supply chain enterprises?

To solve the problems, we construct a three-party stochastic evolutionary game model between financial institutions, core enterprises, and MSMEs and transform blockchain features into parameters to be added to the mathematical model. We introduce stochastic disturbance factors and quantitatively analyze the impact of changes in key parameters on the strategy choice of game players through numerical simulations to explore the evolutionary game path of the decision makers in supply chain finance. We find that the information-sharing incentive factor promotes financial institutions to choose to connect to the blockchain platform, while the information-sharing risk factor and regulatory effort play the opposite role in blockchain platform building. Meanwhile, the allocation of incentive shares significantly impacts core enterprises. We further give the priority and direction of regulation for the adjustment of relevant parameters to provide suggestions for financial institutions to regulate the financing process more effectively.

The rest of the paper is organized as follows. Section 2 presents a review of the related literature. Section 3 constructs a three-party evolutionary game model for accounts receivable financing under supply chain finance with blockchain technology. Section 4 introduces stochastic factors, constructs a three-party stochastic evolutionary game model, and conducts stability analysis on the evolutionary equilibrium solution of the model. Section 5 performs a stochastic Taylor expansion of the model and parametric analysis of the model using numerical simulations. Section 6 summarizes our conclusions.

## 2. Literature Review

The emergence and development of supply chain finance have effectively solved the problem of difficult financing for small and medium-sized enterprises [5]. In 2000, Timme and Williams-Timme [6] first proposed the concept of supply chain finance, which is a financial instrument led by core enterprises to provide a guarantee for upstream and downstream enterprises to obtain external financing through their credit. On this basis, Lamoureux [7] redefined supply chain finance as a process of systematic optimization of the availability and cost of capital, led by the core enterprises. In 2004, Lambert [8] proposed a receivables financing model based on upstream suppliers. However, in the actual financing process, due to the redundant network of upstream and downstream enterprises in the supply chain, collusive fraud is hidden and difficult to investigate [9]. The occurrence of several collusive frauds and defaults indicates that there are incalculable hidden risks in the pledge financing model of accounts receivable. Some scholars believed that information asymmetry is the most important risk faced by supply chain finance [10,11]. Gatteschi et al. [12] argued that blockchain's smart contract technology can effectively improve the efficiency of financial institutions and core enterprises in providing guarantees and discounting to financiers. Some scholars pointed out that blockchain technology features can effectively fit in with the development of supply chain finance [13–16]. Zhou and Liu [17], through a bibliometric data-driven analysis technique, found that the blockchain technique has substantial applications in the field of cross-border e-commerce supply chain, whose contributions mainly focus on a cross-border e-commerce platform, supply chain operations, and data governance and information management. While Hofmann [18] analyzed the characteristics of blockchain technology that can help financial activities, he also argued that its tamper-proof nature can cause certain hidden dangers.

The supply chain finance market is characterized by a distinctly game-like nature. Scholars have conducted a large number of game studies in the context of supply chain finance. Bu and Li [19] constructed a two-subject evolutionary game study between fintech companies and regulatory authorities and found that the strategy choice of fintech companies was influenced by factors, such as extra profits for non-compliant innovation, incentives, and penalties for compliant innovation. The strategy choice of regulatory authorities was influenced by regulatory costs, social evaluation, and negative externalities. Wang et al. [20] constructed a three-party evolutionary game among financial institutions, core enterprises, and MSMEs, analyzed the equilibrium points of the three-party dynamic evolutionary model and its asymptotic stability, and found that the strategy choice of MSMEs was influenced by the cost of default and expected benefits. Zhang [21] analyzed the supply chain financial service model and blockchain incentive mechanism, respectively, from the perspective of the game and gave targeted solutions and countermeasure suggestions for the development of blockchain-driven supply chain financial innovation. Using game theory ideas, Deng and Li [22] firstly modeled and analyzed the automated execution mechanism of smart contracts from the perspective of blockchain node activity technology. In addition, they analyzed the three-party game of supply chain factoring financing process, considering the influence of blockchain from the perspective of supply chain business subjects' decision making.

In earlier studies, some scholars have already captured the existence of high uncertainty in the objective world and taken it into account in the study of game theory [23,24]. Xu et al. [25] used stochastic evolutionary game models to study the stability of the behavior of strategic alliance subjects and solved the sufficient conditions for the alliance to remain stable and disintegrate, respectively, as a way to explain the effectiveness of strategic alliances. Miao et al. [26] also applied the *Itô*'s stochastic differential equation containing white noise in studying the game behavior of owners and contractors in major engineering projects and provided relevant decision-making suggestions for the risk management of interest subjects in major engineering projects. Li et al. [27] built a three-party stochastic evolutionary game model of local government, reporting enterprises, and intermediaries around the phenomenon of R&D manipulation in the declaration of high-tech enterprises,

used *Itô*'s stochastic differential equation theory to analyze the stability conditions of the behavior of game subjects, and quantitatively analyzed the influence of key variables on the subjects' strategy choice.

From the existing literature, firstly, most of the research on the accounts receivable financing model is based on the traditional accounts receivable financing model, and there is less research on the accounts receivable financing model based on supply chain finance. It is mostly two-entity game research, and there is less research on three-party evolution games. On this basis, we have constructed a three-party evolutionary game model for analysis. Secondly, the supply chain finance model proposed by scholars has not departed from the foundation of traditional supply chain finance. Most of the literature studies the combination of blockchain and supply chain finance from the qualitative perspective and application design perspective, and fewer studies explore the application of blockchain technology from the perspective of quantitative analysis. We transform blockchain features into parameters to join the model and consider its impact on the game subjects. Finally, most of the literature ignores the impact of external factors' interference on the strategies of game subjects and cannot reflect the impact of realistic stochastic interference factors on the decisions of each participating subject. We introduce random disturbance factors to make the evolutionary process close to reality.

## 3. Evolutionary Game Model

The traditional supply chain finance receivables financing model is shown in Figure 1. In the receivables financing model of supply chain finance, MSMEs are upstream suppliers, and core enterprises usually purchase from MSMEs in the form of credit. At this time, MSMEs, which mainly rely on cash to maintain their normal operations, need funds to purchase raw materials but are unable to obtain financing from financial institutions due to factors such as small scale, few collateralizable assets, few free assets, weak risk resistance, and low credit rating. After reaching a deal, the core enterprise issues an accounts receivable document to the MSME, which pledges the document to the financial institution. If the core enterprise agrees to use its creditworthiness to guarantee the financing for the MSME, it submits proof of the accounts receivable document and a payment guarantee to the financial institution, which receives it and finances the MSME. At this point, the claims on the receivables are transferred from the MSME to the financial institution, and the core enterprise needs to repay the receivables by the due date.

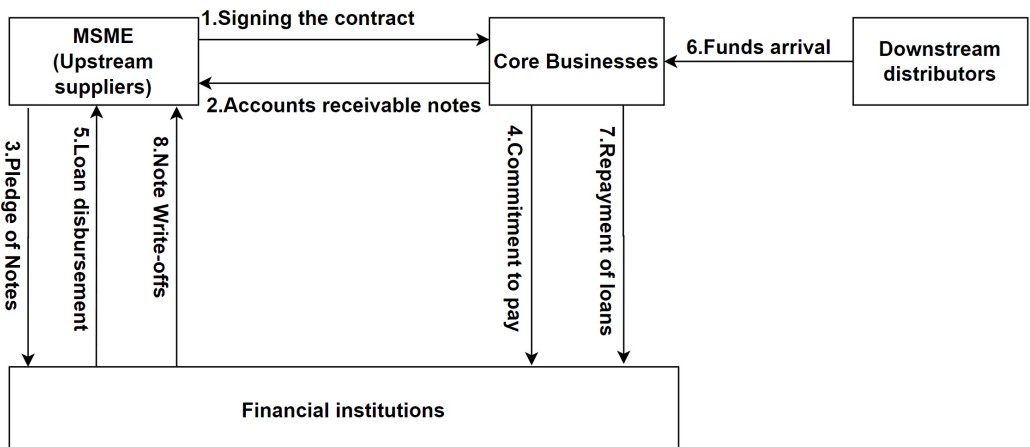

**Figure 1.** Traditional supply chain finance accounts receivable financing model.

The supply chain finance receivables financing model under blockchain is shown in Figure 2. In the actual transaction process, core enterprises and MSMEs may falsify transaction information and accounts receivable to financial institutions for fraudulent lending, and the successful fraudulent lending enterprises may obtain additional spec-

ulative benefits or even run away with the money. Financial institutions need to audit the authenticity of transactions and accounts receivable, but due to the numerous supply chains, varying degrees of information technology, and cumbersome audit processes, financial institutions have to pay high audit costs and spend a lot of time. After the introduction of blockchain technology, docking to the blockchain platform will significantly improve the efficiency of financing. The blockchain consensus algorithm makes the data on the chain self-timestamped. Even if the data are tampered with, there is still a trace, avoiding asymmetry of information from multiple parties. Therefore, in the transaction process, the core enterprises cannot falsify the transaction information after the introduction of blockchain.

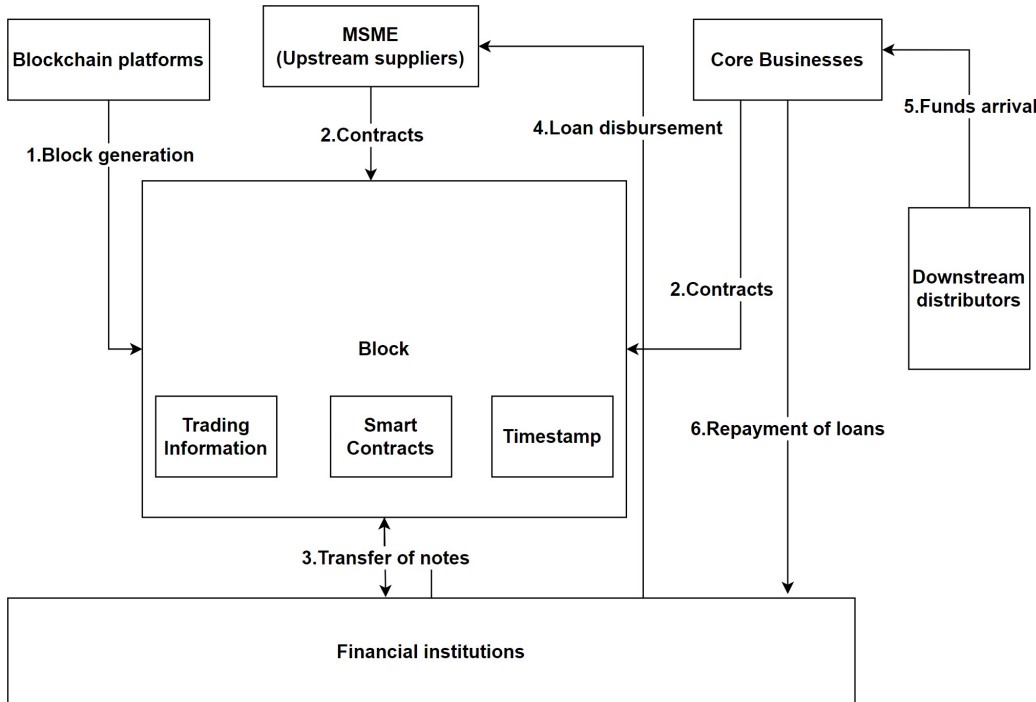

**Figure 2.** Supply chain finance receivables financing model under blockchain.

*3.1. Model Hypothesis*

**Hypothesis 1 (H1).** *Financial institutions, core enterprises, and MSMEs are all finite rational decision makers who learn to maximize their own utility in the game process.*

**Hypothesis 2 (H2).** *The information shared on the blockchain by the participants in the game is real information.*

**Hypothesis 3 (H3).** *Information on the blockchain cannot be tampered with; thus, core and MSMEs cannot conspire to defraud and falsify accounts receivable.*

*3.2. Analysis of the Game Model*

Consider a game model consisting of a financial institution, a core enterprise, and an MSME in a receivables financing model under supply chain finance. In this supply chain finance model, the core enterprise acts as the financing initiator and provides guarantees for the MSMEs, the financial institution provides loans, and the core enterprise is the final repayer. The blockchain platform built by financial institutions can significantly reduce the audit cost of financial institutions, reduce the risk of business default, and improve the efficiency of financing. However, financial institutions may not choose to dock to a blockchain platform because of the high upfront investment costs. As the ultimate repayer, the core enterprises may not repay the loan due to financial constraints or seeking investment interests, on the one hand. On the other hand, it is worried about the pressure of public opinion and penalties from financial institutions if its non-payment behavior is exposed or

investigated. After obtaining financing, MSMEs may, on the one hand, choose to default and use the funds for investment to obtain additional returns due to insufficient production capacity, but it will face penalties and media exposure from financial institutions and core enterprises. On the other hand, MSMEs will use the funds to purchase raw materials for normal production activities and maintain the normal operation of the supply chain.

The optional strategies for each of the three-party game players are: the strategy for the financial institution is {non-docking, docking} with probability $x, 1 - x$; the strategy set for the core enterprise is {non-repayment, repayment} with probability $y, 1 - y$; and the strategy set for the MSME is {default, keep promise} with probability $z, 1 - z$.

Assume that the loan amount of the MSME is $A$, and the interest rate of the loan is $r$. Before docking to the blockchain platform, for a financial institution, the regulatory cost of the financial institution is $\gamma c_1$, and $\gamma$ is the regulatory effort factor. The financial institution's gain from handling pledged receivables financing is $Ar$, and the loss due to the core enterprise's non-repayment is $A(1 + r)$. Since the blockchain is not used, the core enterprise and the MSME conspire to falsify the receivables, the fraud success rate is $1 - \gamma$, and the loss caused by the falsified receivables to the financial institution is denoted by $R_m$. For core enterprises, the gain from financing through pledging accounts receivable is $W_1$, the gain from reinvestment without repayment is $W_2$, the additional gain from colluding to falsify accounts receivable for investment is $W_m$, the penalty from the financial institution for non-repayment is $p_1$, the penalty from the financial institution for falsifying accounts receivable is $p_2$, and the penalty goes to the financial institution. For the MSME, the gain from financing through accounts receivable is $S_1$, the gain from defaulting to reinvest is $S_2$, the additional gain from colluding to falsify accounts receivable for investment is $S_m$, the penalty from the financial institution for its default is $p_3$, the penalty from falsifying accounts receivable is $p_4$, and the penalty goes to the financial institution and the core business, respectively. If the MSME chooses to keep its credit and the core enterprise chooses to repay, both receive an incentive V from the financial institution, of which the core enterprise's share is $\beta_1$. If one party defaults, the incentive goes to the other party.

The cost of building a blockchain platform for financial institutions is $C$. After docking the blockchain platform, the cost of using the blockchain platform is $C$. The core enterprise and the MSME cannot conspire to falsify the receivables. If the core enterprise repays on time and the MSME keeps its promise, they jointly receive a blockchain incentive $G$, of which the core enterprise accounts for $\beta_2$. If one party defaults, the incentive goes to the other party. After docking, $K_1$, $K_2$, and $K_3$ indicate the shared information absorption capacity of a financial institution, core enterprise, and MSME, respectively, and $Q_1$, $Q_2$, and $Q_3$ indicate the information sharing volume of financial institutions, core enterprise, and MSME, respectively, such as loan product information, enterprise credit, warehouse receipt information, etc. Meanwhile, information sharing may trigger risks; $\delta$ denotes the risk coefficient of information sharing, and $\alpha$ denotes the incentive coefficient of information sharing.

### 3.3. Analysis of Payoff Matrix

The scenarios of the different strategic choices of the three subjects are analyzed in the context of the constructed tripartite game scenario.

When the financial institution chooses to dock, the core enterprise chooses to repay, and the MSME chooses to keep its promise, there are no risk costs and benefits arising from fraud and default for the three parties. Therefore, the financial institution's return is $Ar + (\alpha - \delta)Q_1 + K_1(Q_2 + Q_3) - C$, the core enterprise's return is $W_1 + \beta_1 V + \beta_2 G + (\alpha - \delta)Q_2 + K_2(Q_1 + Q_3) - Ar$, and the MSME's return is $S_1 + (1 - \beta_1)V + (1 - \beta_2)G + (\alpha - \delta)Q_3 + K_3(Q_1 + Q_2)$.

When the financial institution chooses to dock, the core enterprise chooses to repay, and the MSME chooses to default, the MSME will be penalized for default and will not receive the blockchain incentive or the trustworthiness incentive, but it will receive the investment return. The core enterprise still needs to repay the loan without receiving the goods and cannot return the funds, cannot obtain the financing income, and obtains all

the trustworthiness incentives and blockchain incentives. Thus, the financial institution's return is $Ar + (\alpha - \delta)Q_1 + K_1(Q_2 + Q_3) - C + p_3$, the core enterprise's return is $V + G + (\alpha - \delta)Q_2 + K_2(Q_1 + Q_3) - A(1 + r)$, and the MSME's return is $S_2 + (\alpha - \delta)Q_3 + K_3(Q_1 + Q_2) - p_3$.

When the financial institution chooses to dock, the core enterprise chooses not to repay, and the MSME chooses to keep its promise, the core enterprise defaults and is penalized for the default and does not receive the blockchain incentive or the trust incentive, but it receives the investment return. The MSME receives the financing proceeds and receives the full compliance incentive and the blockchain incentive. The financial institution cannot recover the loan, so the financial institution's return is $p_1 + (\alpha - \delta)Q_1 + K_1(Q_2 + Q_3) - A(1 + r) - C$, the core enterprise's return is $W_2 + (\alpha - \delta)Q_2 + K_2(Q_1 + Q_3) - p_1$, and the MSME's return is $S_1 + V + G + (\alpha - \delta)Q_3 + K_3(Q_1 + Q_2)$.

When the financial institution chooses to dock, the core enterprise chooses not to repay, and the MSME chooses to default, the financial institution is unable to recover the loan, and the core enterprise is unable to carry out productive activities to return funds for reinvestment because the goods have not been received. Thus, the financial institution's return is $p_1 + p_3 + (\alpha - \delta)Q_1 + K_1(Q_2 + Q_3) - A(1 + r) - C$, the core enterprise's return is $(\alpha - \delta)Q_2 + K_2(Q_1 + Q_3) - p_1$, and the MSME's return is $S_2 + (\alpha - \delta)Q_3 + K_3(Q_1 + Q_2) - p_3$.

When the financial institution chooses not to dock, the core enterprise chooses to repay and MSME chooses to keep its promise, they are unable to obtain information-sharing gains and blockchain incentives from the blockchain platform. Core enterprises and MSMEs can fraudulently obtain loans from financial institutions by falsifying their accounts receivable, and financial institutions penalize them regardless of whether the fraud succeeds or fails. The financial institution's return is $Ar + p_2 + p_4 - \gamma c_1 - (1 - \gamma)R_m$, the core enterprise's return is $W_1 + \beta_1 V + (1 - \gamma)W_m - p_2 - Ar$, and the MSME's return is $S_1 + (1 - \beta_1)V + (1 - \gamma)S_m - p_4$.

When the financial institution chooses not to dock, the core enterprise chooses to repay the loan, and the MSME chooses to default, the MSME will be penalized for default and will not receive the blockchain incentive as well as the trustworthiness incentive, but it receives the investment return. The core enterprise still needs to repay the loan without receiving the goods, cannot return the funds, and cannot obtain the financing benefit and obtain the full trustworthiness incentive. Thus, the financial institution's return is $Ar + p_2 + p_4 + p_3 - \gamma c_1 - (1 - \gamma)R_m$, the core enterprise's return is $W_1 + V + (1 - \gamma)W_m - p_2 - A(1 + r)$, and the MSME's return is $S_2 + (1 - \gamma)S_m - p_3 - p_4$.

When the financial institution chooses not to dock, the core enterprise chooses not to repay, and the MSME chooses to keep the loan, the financial institution cannot recover the loan. Therefore, the financial institution's return is $p_1 + p_2 + p_4 - A(1 + r) - \gamma c_1 - (1 - \gamma)R_m$, the core enterprise's return is $W_2 + (1 - \gamma)W_m - p_1 - p_2$, and the MSME's return is $S_1 + V + (1 - \gamma)S_m - p_4$.

When the financial institution chooses not to dock, the core enterprise chooses not to repay, and the MSME chooses to default, the financial institution's return is $p_1 + p_2 + p_3 + p_4 - A(1 + r) - \gamma c_1 - (1 - \gamma)R_m$, the core enterprise's return is $(1 - \gamma)W_m - p_1 - p_2$, and the MSME's return is $S_2 + (1 - \gamma)S_m - p_3 - p_4$. The payoff matrix of financial institutions, core enterprises, and MSMEs is shown in Table 1.

**Table 1.** Payoff matrix of financial institutions, core enterprises, and MSMEs.

| Strategic | Financial Institutions | Core Enterprises | MSMEs |
|---|---|---|---|
| (docking, repayment, keep promise) | $Ar + (\alpha - \delta)Q_1 + K_1(Q_2 + Q_3) - C$ | $W_1 + \beta_1 V + \beta_2 G + (\alpha - \delta)Q_2 + K_2(Q_1 + Q_3) - Ar$ | $S_1 + (1 - \beta_1)V + (1 - \beta_2)G + (\alpha - \delta)Q_3 + K_3(Q_1 + Q_2)$ |
| (docking, repayment, default) | $Ar + (\alpha - \delta)Q_1 + K_1(Q_2 + Q_3) - C + p_3$ | $V + G + (\alpha - \delta)Q_2 + K_2(Q_1 + Q_3) - A(1 + r)$ | $S_2 + (\alpha - \delta)Q_3 + K_3(Q_1 + Q_2) - p_3$ |
| (docking, non-repayment, keep promise) | $p_1 + (\alpha - \delta)Q_1 + K_1(Q_2 + Q_3) - A(1 + r) - C$ | $W_2 + (\alpha - \delta)Q_2 + K_2(Q_1 + Q_3) - p_1$ | $S_1 + V + G + (\alpha - \delta)Q_3 + K_3(Q_1 + Q_2)$ |
| (docking, non-repayment, default) | $p_1 + p_3 + (\alpha - \delta)Q_1 + K_1(Q_2 + Q_3) - A(1 + r) - C$ | $(\alpha - \delta)Q_2 + K_2(Q_1 + Q_3) - p_1$ | $S_2 + (\alpha - \delta)Q_3 + K_3(Q_1 + Q_2) - p_3$ |
| (non-docking, repayment, keep promise) | $Ar + p_2 + p_4 - \gamma c_1 - (1 - \gamma)R_m$ | $W_1 + \beta_1 V + (1 - \gamma)W_m - p_2 - Ar$ | $S_1 + (1 - \beta_1)V + (1 - \gamma)S_m - p_4$ |
| (non-docking, repayment, default) | $Ar + p_2 + p_4 + p_3 - \gamma c_1 - (1 - \gamma)R_m$ | $V + (1 - \gamma)W_m - p_2 - A(1 + r)$ | $S_2 + (1 - \gamma)S_m - p_3 - p_4$ |
| (nondocking, nonrepayment, keep promise) | $p_1 + p_2 + p_4 - A(1 + r) - \gamma c_1 - (1 - \gamma)R_m$ | $W_2 + (1 - \gamma)W_m - p_1 - p_2$ | $S_1 + V + (1 - \gamma)S_m - p_4$ |
| (non-docking, non-repayment, default) | $p_1 + p_2 + p_3 + p_4 - A(1 + r) - \gamma c_1 - (1 - \gamma)R_m$ | $(1 - \gamma)W_m - p_1 - p_2$ | $S_2 + (1 - \gamma)S_m - p_3 - p_4$ |

### 3.4. Replication of Dynamic Equations

Assuming that the expected return of a financial institution adopting a "non-docking" or "docking" strategy is $U_{11}$ and $U_{12}$, respectively, the average expected return of a financial institution is $\overline{U}_1$, with:

$$
\begin{aligned}
U_{11} =& yz[p_1 + p_2 + p_2 + p_3 + p_4 - A(1 + r) - \gamma c_1 - (1 - \gamma)R_m] \\
&+ y(1 - z)[p_1 + p_2 + p_4 - A(1 + r) - \gamma c_1 - (1 - \gamma)R_m] \\
&+ (1 - y)z[Ar + p_2 + p_3 + p_4 - \gamma c_1 - (1 - \gamma)R_m] \\
&+ (1 - y)(1 - z)[Ar + p_2 + p_4 - \gamma c_1 - (1 - \gamma)R_m] \\
U_{12} =& yz[p_1 + p_2 + p_3 + (\alpha - \delta)Q_1 + K_1(Q_2 + Q_3) - A(1 - r) - C] \\
&+ y(1 - z)[p_1 + (\alpha - \delta)Q_1 + K_1(Q_2 + Q_3) - A(1 - r) - C] \\
&+ (1 - y)z[Ar + p_3 + (\alpha - \delta)Q_1 + K_1(Q_2 + Q_3) - C] \\
&+ (1 - y)(1 - z)[Ar + (\alpha - \delta)Q_1 + K_1(Q_2 + Q_3) - C] \\
\overline{U}_1 =& xU_{11} + (1 - x)U_{12},
\end{aligned} \tag{1}
$$

The replication dynamic equation for the financial institution is obtained as follows:

$$
\begin{aligned}
\frac{dx}{dt} &= x(U_{11} - \overline{U}_1) \\
&= x(1 - x)[C + p_2 + p_4 - \gamma c_1 - (1 - \gamma)R_m - (\alpha - \delta)Q_1 - K_1(Q_2 + Q_3)].
\end{aligned} \tag{2}
$$

Assuming that the expected returns of the core enterprise adopting the "non-repayment" or "repayment" strategy are $U_{21}$ and $U_{22}$, respectively, the average expected return of the core enterprise is $\overline{U}_2$, with:

$$
\begin{aligned}
U_{21} =& xz[(1 - \gamma)W_m - p_1 - p_2] \\
&+ x(1 - z)[W_2 + (1 - \gamma)W_m - p_1 - p_2] \\
&+ (1 - x)z[(\alpha - \delta)Q_1 + K_2(Q_1 + Q_3)] \\
&+ (1 - x)(1 - z)[W_2 + (\alpha - \delta)Q_1 + K_2(Q_1 + Q_3) - p_1] \\
U_{22} =& xz[V + (1 - \gamma)W_m - p_2 - A(1 - r)] \\
&+ x(1 - z)[W_1 + \beta_1 V + (1 - \gamma)W_m - p_2 - Ar] \\
&+ (1 - x)z[V + G + (\alpha - \delta)Q_1 + K_21(Q_1 + Q_3) - A(1 - r)] \\
&+ (1 - x)(1 - z)[W_1 + \beta_1 V + \beta_2 G + (\alpha - \delta)Q_1 + K_2(Q_1 + Q_3) - Ar] \\
\overline{U}_2 =& yU_{21} + (1 - y)U_{22},
\end{aligned} \tag{3}
$$

The replication dynamic equation for the core enterprise is obtained as follows:

$$
\begin{aligned}
\frac{dy}{dt} =& y(U_{21} - \overline{U}_2) \\
=& y(1-y)[(1-z)(W_2 - W_1 - \beta_1 V) + z(A-V) + Ar \\
& - p_1 - (1-x)zG - (1-x)(1-z)\beta_2 G].
\end{aligned}
\tag{4}
$$

Assuming that the expected returns of MSMEs adopting the "default" or "keep promise" strategy are $U_{31}$ and $U_{32}$, respectively, the average expected return for MSMEs is $\overline{U}_3$, with:

$$
\begin{aligned}
U_{31} =& xy[S_2 + (1-\gamma)S_m - p_3 - p_4] \\
& + x(1-y)[S_2 + (1-\gamma)S_m - p_3 - p_4] \\
& + (1-x)y[S_2 + (\alpha-\delta)Q_1 + K_3(Q_1 + Q_2) - p_3] \\
& + (1-x)(1-y)[S_2 + (\alpha-\delta)Q_1 + K_3(Q_1 + Q_2) - p_3] \\
U_{32} =& xy[S_1 + V + (1-\gamma)S_m - p_4] \\
& + x(1-y)[S_1 + (1-\beta_1)V + (1-\gamma)S_m - p_4] \\
& + (1-x)y[S_1 + (\alpha-\delta)Q_1 + K_3(Q_1 + Q_2) \\
& + (1-x)(1-y)[S_1 + (1-\beta_1)V + (1-\beta_2)G + (\alpha-\delta)Q_1 + K_3(Q_1 + Q_2)] \\
\overline{U}_3 =& zU_{31} + (1-z)U_{32},
\end{aligned}
\tag{5}
$$

The replication dynamics equation for MSMEs is obtained as follows:

$$
\begin{aligned}
\frac{dz}{dt} =& z(U_{31} - \overline{U}_3) \\
=& z(1-z)[S_2 - p_3 - S_1 - V + (1-y)\beta_1 V - (1-x)G + (1-x)(1-y)\beta_2 G].
\end{aligned}
\tag{6}
$$

Since $1-x$, $1-y$, and $1-z$ are non-negative and do not affect the outcome of the evolutionary game, we transform the replication dynamic equation into the following form:

$$
\begin{aligned}
dx =& x[C + p_2 + p_4 - \gamma c_1 - (1-\gamma)R_m - (\alpha-\delta)Q_1 - K_1(Q_2 + Q_3)]dt \\
dy =& y[(1-z)(W_2 - W_1 - \beta_1 V) + z(A-V) + Ar - p_1 - (1-x)zG - (1-x)(1-z)\beta_2 G]dt \\
dz =& z(1-z)[S_2 - p_3 - S_1 - V + (1-y)\beta_1 V - (1-x)G + (1-x)(1-y)\beta_2 G]dt.
\end{aligned}
\tag{7}
$$

## 4. Construction of a Stochastic Evolutionary Game Model

### 4.1. Random Factors Perturbing the Three-Party Game

During the operation of supply chain finance, the core enterprise, the MSME, and the financial institution face greater uncertainty. On the one hand, in addition to the production and operating conditions of enterprises, the changing sentiment and risk appetite of each group will directly influence the strategies of participants. On the other hand, external factors, such as changes in the supply chain financial system and macroeconomic policies, will also interfere with the behavior of participants. Therefore, the original evolutionary game cannot truly reflect the strategic adjustment process of each participant. To overcome this shortcoming, we add stochastic factors to the evolutionary game model. Gaussian white noise is introduced into the replicated dynamic equations as follows:

$$
dx(t) = x(t)[C + p_2 + p_4 - \gamma c_1 - (1-\gamma)R_m - (\alpha-\delta)Q_1 - K_1(Q_2 + Q_3)]dt
\tag{8}
$$

$$
\begin{aligned}
dy(t) =& y(t)[(1-z)(W_2 - W_1 - \beta_1 V) + z(A-V) + Ar - p_1 - (1-x)zG \\
& - (1-x)(1-z)\beta_2 G]dt + \sigma y(t)d\omega(t)
\end{aligned}
\tag{9}
$$

$$
\begin{aligned}
dz(t) =& z(t)[S_2 - p_3 - S_1 - V + (1-y)\beta_1 V - (1-x)G + (1-x)(1-y)\beta_2 G]dt \\
& + \sigma z(t)d\omega(t)
\end{aligned}
\tag{10}
$$

In Equations (8)–(10), $\sigma$ denotes the strength of the random disturbance, and $\omega(t)$ denotes the one-dimensional standard Brownian motion. Brownian motion is a stochastic fluctuation phenomenon that captures the effect of random disturbances on the subject of the game well. Gaussian white noise is $d\omega(t)$. When $t > 0$ and the time step $h > 0$, $\omega(t + h) - \omega(t) \sim N(0, h)$.

### 4.2. Stability Analysis of Evolutionary Equilibrium Solutions

Assuming an initial state of $t = 0$ and initial values of $x(0) = 0$, $y(0) = 0$, and $z(0) = 0$, we can find $x(t) = 0$, $y(t) = 0$, and $z(t) = 0$, which are equilibrium solutions to replicate the dynamic equations. In other words, in the absence of external interference, the system will always remain at the point where the financial institution chooses "docking", the core enterprise chooses "repay", and the MSME chooses "keep promise". However, due to the high level of uncertainty in the real world, this ideal situation is usually impossible to achieve, and each player in the game is subject to more or less random factors. Therefore, it is necessary to consider the impact of stochastic disturbances on the stability of the system [28].

**Theorem 1.** *For the stochastic differential equation is shown below:*

$$dx(t) = f(t, x(t))dt + g(t, x(t))d\omega(t), x(t_0) = x_0, \tag{11}$$

*let $x(t) = x(t, x_0)$ be a solution to the equation given a continuously differentiable function $V(t, x)$ and positive numbers; there exists $c_1|x|^p \leq c_2|x|^p$, $t \geq 0$. Let $LV(t, x) = V_t(t, x) + V_x(t, x)f(t, x) + 1/2g^2(t, x)V_{xx}(t, x)$: (1) If there is a positive number $\gamma$ which satisfies $LV(t, x) \leq -\gamma V(t, x)$, $t \geq 0$, then the zero solution of the equation with p-th moment exponent is stable and $E|x(t, x_0)|^p < (c_2/c_1)|x_0|^p e^{-\gamma t}$, $t \geq 0$. (2) If there is a positive number $\gamma$ which satisfies $LV(t, x) \geq \gamma V(t, x)$, $t \geq 0$, then the zero solution p-th moment of the equation is unstable and $E|x(t, x_0)|^p \geq (c_2/c_1)|x_0|^p e^{-\gamma t}$, $t \geq 0$.*

For the replicated dynamic Equations (8)–(10) of the stochastic evolutionary game we have created, let $(t, x) = x, V(t, y) = y, V(t, z) = z, c_1 = 1, c_2 = 1, p = 1$ and $\gamma = 1$; $LV(t, x) = f(t, x), LV(t, y) = f(t, y), LV(t, z) = f(t, z)$. If the zero solution of the equation with $p$-th moment exponential is stable, it needs to satisfy:

$$\begin{cases} x[C + p_2 + p_4 - \gamma c_1 - (1 - \gamma)R_m - (\alpha - \delta)Q_1 - K_1(Q_2 + Q_3)] \leq -x \\ y[(1 - z)(W_2 - W_1 - \beta_1 V) + z(A - V) + Ar - p_1 - (1 - x)zG - (1 - x)(1 - z)\beta_2 G] \leq -y \\ z[S_2 - p_3 - S_1 - V + (1 - y)\beta_1 V - (1 - x)G + (1 - x)(1 - y)\beta_2 G] \leq -z \end{cases} \tag{12}$$

Since $x, y, z \in [0, 1]$, the conditions for satisfying the above equation are:

$$\begin{cases} [C + p_2 + p_4 - \gamma c_1 - (1 - \gamma)R_m - (\alpha - \delta)Q_1 - K_1(Q_2 + Q_3)]x \leq -x \\ [(1 - z)(W_2 - W_1 - \beta_1 V) + z(A - V) + Ar - p_1 - (1 - x)zG - (1 - x)(1 - z)\beta_2 G]y \leq -y \\ [S_2 - p_3 - S_1 - V + (1 - y)\beta_1 V - (1 - x)G + (1 - x)(1 - y)\beta_2 G]z \leq -z \end{cases} \tag{13}$$

The zero solution $p$-th moment exponential is stable when all three conditions are met. This means that over time the proportion of non-cooperative strategies (financial institutions choosing "non-docking", core enterprises choosing "non-repayment", and MSMEs choosing "default") will decay exponentially to zero. At this point, the only evolving and stable strategy is the cooperative strategy (financial institutions choose "docking", core enterprises choose "repayment", and MSMEs choose "keep promise").

## 5. Numerical Simulation and Analysis

### 5.1. Stochastic Taylor Expansions of Replicating Dynamic Equation

Since the replica dynamic equations are nonlinear *Itô*'s stochastic differential equations for which analytical solutions are not directly available, the equations are solved numer-

ically using the stochastic Taylor expansion. The following *Itô*'s stochastic differential equations are first discussed:

$$x(t) = f(t, x(t))dt + g(t, x(t))d\omega(t). \tag{14}$$

In this equation, $t \in [t_0, T]$, $x(t_0) = x_0$, $x_0 \in R$ and $\omega(t)$ denotes one-dimensional standard Brownian motion. Let $h = (T - t_0)/N$, $t_n = t_0 + nh$, and the stochastic Taylor expansion of the equation is as follows:

$$
\begin{aligned}
x(t_{n+1}) = &\, x(t_n) + hf(x(t_n)) + \Delta\omega_n g(x(t_n)) + \frac{1}{2}[(\Delta\omega_n)^2 - h]g(x(t_n))g'(x(t_n)) \\
&+ \frac{1}{2}h^2[f(x(t_n))f'(x(t_n))] + \frac{1}{2}g^2(x(t_n))f''(x(t_n))] + R
\end{aligned}
\tag{15}
$$

In the above expansion, R is the residual term. We use the Milstein numerical method to solve the equations [29]. In the Milstein method, the stochastic Taylor expansion is truncated to a first-order expression, and the residual term is discarded. As a result, the equation is modified as follows:

$$x(t_{n+1}) = x(t_n) + hf(x(t_n)) + \Delta\omega_n g(x(t_n)) + \frac{1}{2}\left[(\Delta\omega_n)^2 - h\right]g(x(t_n))g'(x(t_n)). \tag{16}$$

Therefore, using the Milstein method to solve the SDEs (8)–(10), we can obtain:

$$
\begin{aligned}
x(t_{n+1}) = &\, x(t_n) + hx(t_n)[C + p_2 + p_4 - \gamma c_1 - (1 - \gamma)R_m - (\alpha - \delta)Q_1 - K_1(Q_2 + Q_3)] + \Delta\omega_n \sigma x(t_n) \\
&+ \frac{1}{2}[(\Delta\omega_n)^2 - h]\sigma^2 x(t_n)
\end{aligned}
\tag{17}
$$

$$
\begin{aligned}
y(t_{n+1}) = &\, y(t_n) + hy(t_n)[(1 - z(t_n))(W_2 - W_1 - \beta_1 V) + z(t_n)(A - V) + Ar - p_1 - (1 - x(t_n))z(t_n)G \\
&- (1 - x(t_n))(1 - z(t_n))\beta_2 G] + \Delta\omega_n \sigma y(t_n) + \frac{1}{2}[(\Delta\omega_n)^2 - h]\sigma^2 y(t_n)
\end{aligned}
\tag{18}
$$

$$
\begin{aligned}
z(t_{n+1}) = &\, z(t_n) + hz(t_n)[S_2 - p_3 - S_1 - V + (1 - y(t_n))\beta_1 V - (1 - x(t_n))G + (1 - x(t_n))(1 - y(t_n))\beta_2 G] \\
&+ \Delta\omega_n \sigma z(t_n) + \frac{1}{2}[(\Delta\omega_n)^2 - h]\sigma^2 z(t_n)
\end{aligned}
\tag{19}
$$

### 5.2. Parameter Sensitivity Analysis

We use R software to simulate the stochastic evolutionary game model, focusing on the impact of regulatory effort, information-sharing risk coefficient, information-sharing incentive coefficient, and incentive share on the choice of subject strategy. The relevant variables were taken as follows: $A = 6, r = 0.01, c_1 = 2, R_m = 1, W_1 = 11, W_2 = 8, p_1 = 12, p_2 = 2, p_3 = 7, p_4 = 3.5, S_1 = 8, S_2 = 9, V = 20, C = 3, G = 2.5, K_1 = 0.5, Q_1 = 10, Q_2 = 20$, and $Q_3 = 20$. All the initial values of the coefficients were set to 0.5. In the comparative analysis of the correlation coefficients, a low value of 0.1 and a high value of 0.9 are taken as controls.

### 5.2.1. Information Sharing Factor

With other parameters being held constant, the impact of changes in the information-sharing risk factor $\delta$ and the information-sharing incentive factor $\alpha$ on the evolutionary path of strategies for financial institutions, core enterprises, and MSMEs are analyzed. The simulation results are shown in the subplots a–f of Figure 3. Letting $\delta = 0.1$, 0.5, and 0.9, respectively, at $t = 0.03$, for financial institutions, the proportion of financial institutions that choose not to interface with the blockchain platform increases significantly as the information-sharing risk factor increases. The larger the information-sharing risk coefficient, the greater the risk faced by the financial institution and the more inclined the financial institution is not to dock. However, the increase in $\delta$ does not have the same

impact on core and MSMEs. As $\delta$ increases from 0.1 to 0.9, the proportion of core enterprises choosing not to repay decreases, as do MSMEs. Letting $\alpha = 0.1, 0.5$, and 0.9, respectively, at $t = 0.03$, for financial institutions, as the information-sharing incentive factor increases, the proportion of financial institutions choosing not to interface with blockchain platforms changes less, and the impact of changes in the information-sharing incentive factor on financial institutions is insignificant. However, the increase in $\alpha$ does not affect core and MSMEs in the same way. When $\alpha$ increases from 0.1 to 0.9, the proportion of core enterprises choosing not to repay the loan decreases significantly, as do MSMEs.

In terms of the speed of convergence of the strategies, increasing the information-sharing risk factor $\delta$ and the information-sharing incentive factor $\alpha$ from 0.1 to 0.9 has inconsistent effects on financial institutions, core enterprises, and MSMEs. When $\delta = 0.1$, the moment when the proportion of financial institutions adopting the non-docking blockchain platform strategy first reaches 0 is $t = 0.21$, the moment when the proportion of core enterprises adopting the non-repayment strategy first reaches 0 is $t = 0.42$, and the moment when the proportion of MSMEs adopting the default strategy first reaches 0 is $t = 0.29$. When $\delta = 0.9$, the proportion of financial institutions adopting the non-docking blockchain platform strategy at the moment when it first reaches zero is $t = 0.53$, an increase of 0.32, the moment when the proportion of core enterprises adopting a non-repayment strategy first reaches zero is $t = 0.23$, a shortening of 0.19, and the moment when the proportion of MSMEs adopting a default strategy first reaches zero is $t = 0.18$, a shortening of 0.11. Thus, the change in the information-sharing risk factor $\delta$ has a significant impact on the speed at which financial institutions and core and MSMEs reach zero. There is a contradiction between the speed of convergence of the firm strategy to zero. When $\alpha = 0.1$, the moment when the proportion of financial institutions with a non-docking blockchain platform strategy first reaches zero is $t = 0.3$, the moment when the proportion of core enterprises with a non-repayment strategy first reaches zero is $t = 0.42$, and the moment when the proportion of MSMEs with a default strategy first reaches zero is $t = 0.29$. When $\alpha = 0.9$, the moment when the proportion of financial institutions with a non-docking blockchain platform strategy first reaches zero is $t = 0.32$, an increase of 0.02, the moment when the proportion of core enterprises adopting a non-repayment strategy first reaches zero is $t = 0.23$, a shortening of 0.19, and the moment when the proportion of MSMEs adopting a default strategy first reaches zero is $t = 0.18$, a shortening of 0.11. Thus, the change in the incentive coefficient for information-sharing $\alpha$ has a significant impact on the speed at which financial institutions and core, MSMEs reach zero. There is a contradiction between the speed of convergence to zero of the firms' strategies.

In terms of the degree of change in strategy, the proportion of financial institutions choosing not to dock reaches a maximum of 79.61% at $t = 0.03$ when $\delta = 0.9$, but when $\delta = 0.1$, this proportion is 29.91% at moment $t = 0.03$, a decrease of 49.7%. For core enterprises, the proportion choosing not to repay reaches a maximum of 68.71% at moment $t = 0.03$ when $\delta = 0.1$. When $\delta = 0.9$, this proportion drops to 32.89%, a decrease of 38.82%. For MSMEs, the proportion choosing to default reaches a maximum of 55.21% at moment $t = 0.02$ when $\delta = 0.1$, and this proportion drops to 31.07% when $\delta = 0.9$, a decrease of 24.14%. When $\alpha = 0.9$, the proportion of financial institutions choosing not to dovetail reaches a maximum of 63.62% at $t = 0.03$, but when $\delta = 0.1$, this proportion is 39.26% at moment $t = 0.03$, a decrease of 24.36%. For enterprises, the proportion choosing not to repay reaches a maximum of 68.85% at moment $t = 0.03$ when $\alpha = 0.1$. When $\alpha = 0.9$, this proportion drops to 32.82%, a decrease of 36.03%. For MSMEs, the proportion choosing to default reaches a maximum of 55.23% at moment $t = 0.02$ when $\alpha = 0.1$, and this proportion drops to 31.03% when $\alpha = 0.9$, a decrease of 24.2%.

The larger the information-sharing risk factor $\delta$, the more financial institutions tend not to match, the more core enterprises tend to repay, and the more MSMEs tend to keep their credit. The larger the information-sharing incentive coefficient $\alpha$, the more financial institutions tend not to dock, the more core enterprises tend to repay, and the more MSMEs tend to keep their credit. Financial institutions are sensitive to the information-sharing

risk coefficient and less sensitive to the information-sharing incentive coefficient, and core enterprises and MSMEs are equally sensitive to the information-sharing risk and incentive coefficients. This indicates that financial institutions are more concerned about the risk of information leakage they will face when interfacing with blockchain platforms, and they are likely to choose to share less information and more inclined not to interfere with blockchain platforms.

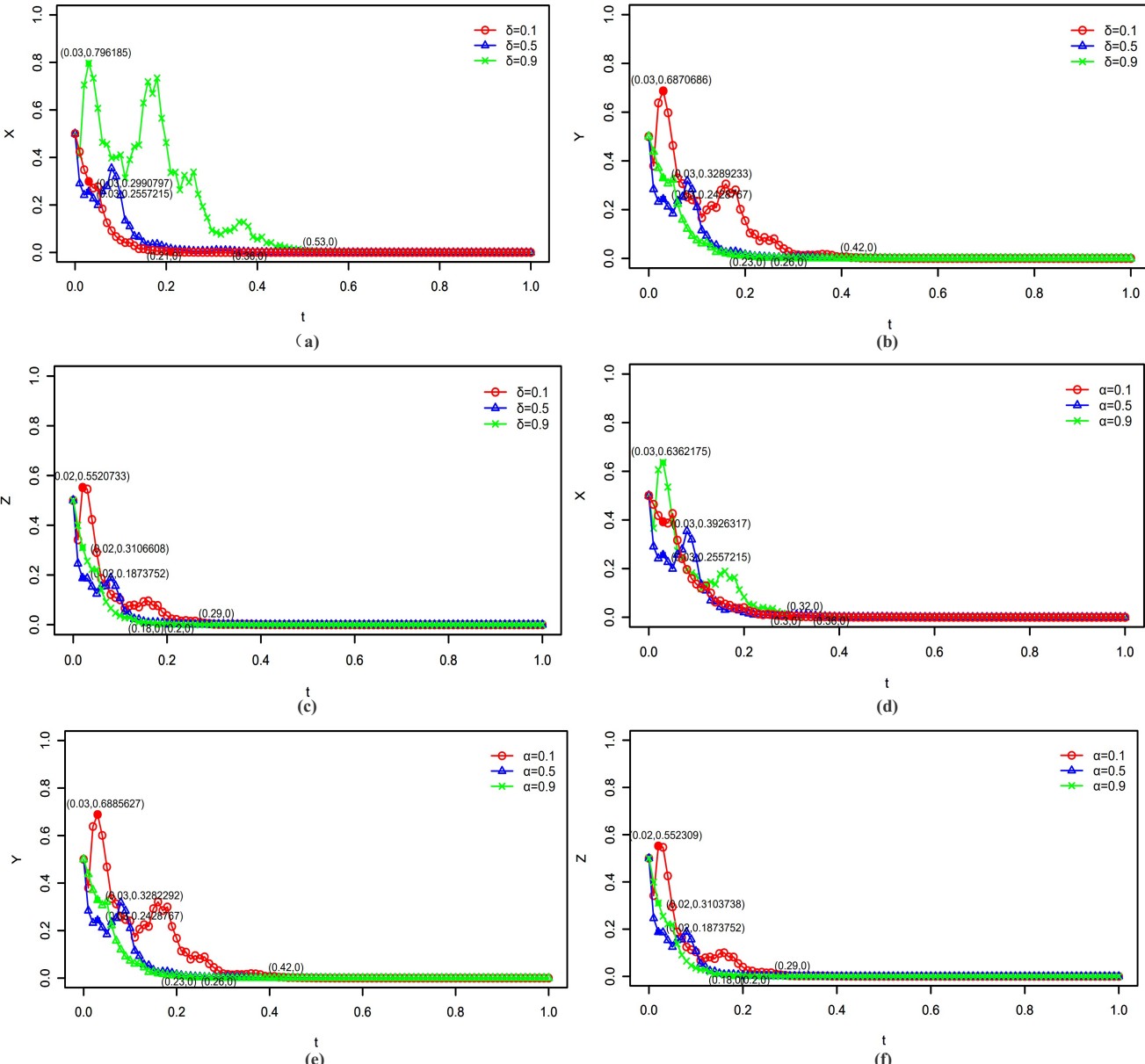

**Figure 3.** The impact of information-sharing coefficients on the gaming strategies of financial institutions, core enterprises, and MSMEs. (**a**) The impact of information-sharing risk factor $\delta$ on the gaming strategies of financial institutions. (**b**) The impact of nformation-sharing risk factor $\delta$ on the gaming strategies of core enterprises. (**c**) The impact information-sharing risk factor $\delta$ on the gaming strategies of MSMEs. (**d**) The impact information-sharing incentive factor $\alpha$ on the gaming strategies of financial institutions. (**e**) The impact information-sharing incentive factor $\alpha$ on the gaming strategies of core enterprises. (**f**) The impact information-sharing incentive factor $\alpha$ on the gaming strategies of MSMEs.

5.2.2. Regulatory Efforts

Holding other parameters constant, we analyze the impact of changes in the regulatory intensity of financial institutions $\gamma$ on the evolutionary path of the strategies of financial institutions, core enterprises, and MSMEs when not docked to the blockchain platform. Let $\gamma = 0.1, 0.5$, and $0.9$, respectively, and the simulation results are shown in the subplots a–c of Figure 4. At $t = 0.03$, for financial institutions, the proportion choosing not to dock to the blockchain platform increases significantly as regulation increases. The increase in $\gamma$ has the opposite effect on core and MSMEs; as $\gamma$ increases from 0.1 to 0.9, the proportion of core enterprises choosing not to repay decreases significantly, as do MSMEs. This may be because when not connected to a blockchain platform, financial institutions strengthen regulation and pay more for regulation, and the less likely core enterprises and MSMEs are to choose to collude in fraud because the expected return from such collusion is reduced. As a result, the expected loss for financial institutions is lower. Therefore, increased regulation can achieve the same effect of reducing fraud and default risk for financial institutions as when connected to a blockchain platform.

In terms of the rate of convergence of the strategy, increasing the regulation $\gamma$ from 0.1 to 0.9 has an inconsistent impact on financial institutions, core enterprises, and MSMEs. When $\gamma = 0.1$, the moment when the proportion of financial institutions with a non-docking blockchain platform strategy first reaches zero is $t = 0.23$, the moment when the proportion of core enterprises with a non-repayment strategy first reaches zero is $t = 0.42$, and the moment when the proportion of MSMEs with a default strategy first reaches zero is $t = 0.42$. When $\gamma = 0.9$, the proportion of financial institutions with a non-docking blockchain platform strategy at the moment when it first reaches zero is $t = 0.29$, an increase of 0.2, the moment when the proportion of core enterprises with a non-repayment strategy first reaches zero is $t = 0.23$, a shortening of 0.19, and the moment when the proportion of MSMEs with a default strategy first reaches zero is $t = 0.18$, a shortening of 0.11. Thus, the change in regulatory intensity $\gamma$ has a significant impact on the convergence of the strategies of financial institutions and core and MSMEs to zero at a rate that is contradictory to each other.

In terms of the degree of change in strategy, the proportion of financial institutions choosing not to dock reaches a maximum of 70.54% at $t = 0.03$ when $\gamma = 0.9$, but when $\gamma = 0.1$, this proportion is 34.84% at moment $t = 0.03$, a decrease of 35.7%. For core enterprises, the proportion choosing not to repay reaches a maximum of 68.79% at moment $t = 0.03$ when $\gamma = 0.1$. When $\gamma = 0.9$, this proportion drops to 32.89%, a decrease of 32.85%. For MSMEs, the proportion choosing to default reaches a maximum of 55.22% at moment $t = 0.02$ when $\gamma = 0.1$, and this proportion drops to 31.05% when $\gamma = 0.9$, a decrease of 24.17%.

Therefore, when not interfacing with a blockchain platform, financial institutions' regulation $\gamma$ has a significant impact on the strategy choice of core and MSMEs, with priority regulation $\gamma$ for core enterprises and second regulation $\gamma$ for MSMEs. When financial institutions have invested enough regulatory costs, they are less inclined to interface with blockchain platforms. Financial institutions need to make a good trade-off between technology and human costs before deciding on the level of regulation. As financial institutions pay higher sunk costs when they do not connect to a blockchain platform, they are less inclined to pay the additional cost of connecting to a blockchain platform, even though connecting to a blockchain platform will reduce this portion of regulatory costs.

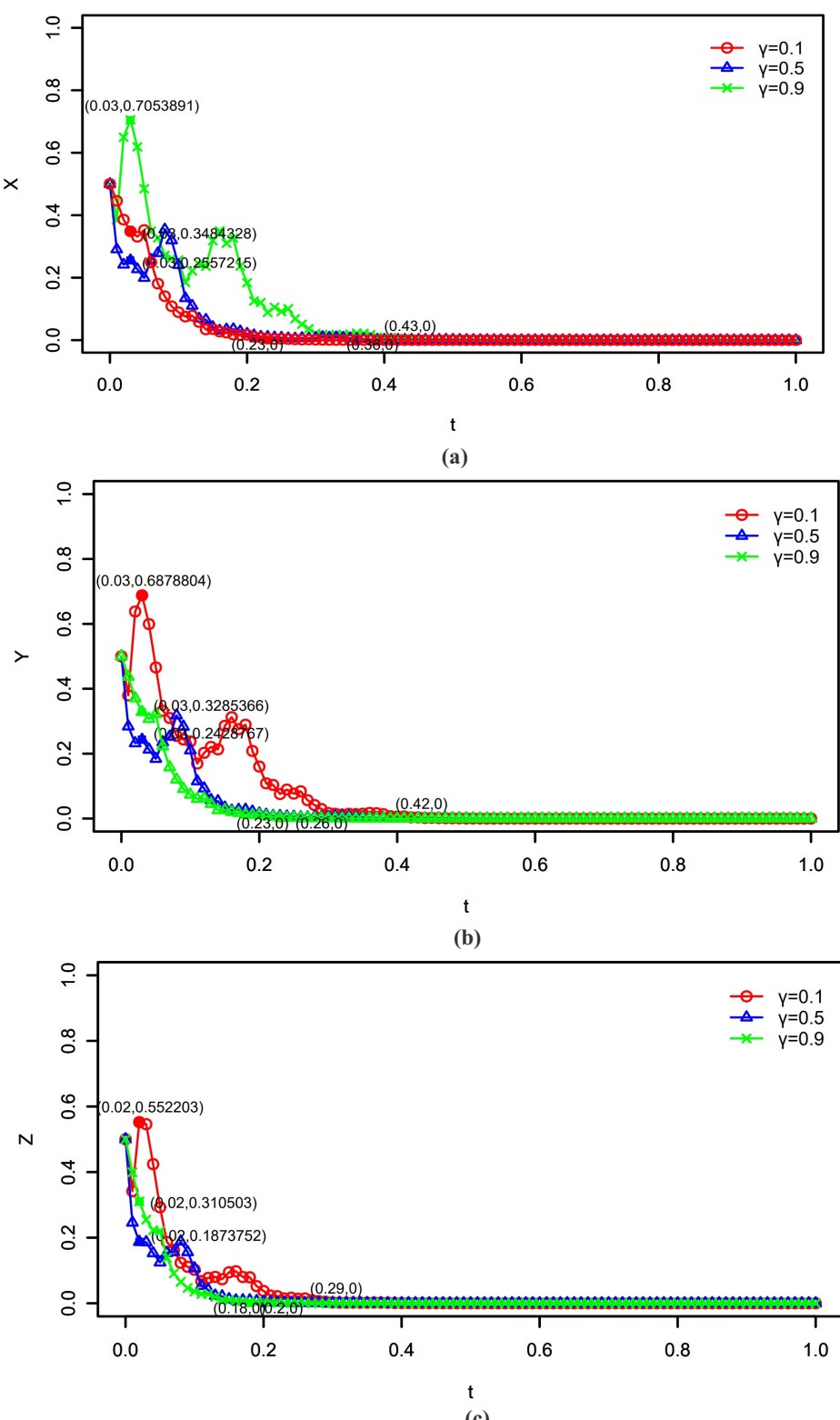

**Figure 4.** The impact of regulatory effort on the choice of strategy of financial institutions, core enterprises, and MSMEs. (**a**) The impact of regulatory effort $\gamma$ on the choice of strategy of financial institutions. (**b**) The impact of regulatory effort $\gamma$ on the choice of strategy of core enterprises. (**c**) The impact of regulatory effort $\gamma$ on the choice of strategy of MSMEs.

### 5.2.3. Incentive Share

Holding other parameters constant, we analyze the impact of the core enterprise share of credit-keeping incentives $\beta_1$ and blockchain incentives $\beta_2$ on the strategy choice of core enterprises and MSMEs. The simulation results are shown in the subplots a–d of Figure 5, respectively. When $\beta_1$ increases from 0.1 to 0.9, the proportion of core enterprises choosing not to repay the loan decreases significantly, while the proportion of MSMEs choosing to default does not change significantly. Let $\beta_2 = 0.1$, 0.5, and 0.9 respectively, when $\beta_2$ increases from 0.1 to 0.9, the proportion of core enterprises choosing not to repay decreases, and the proportion of MSMEs choosing to default decreases, with insignificant changes.

In terms of the speed of convergence of the strategy, increasing the proportion of core enterprises with a trustworthy incentive $\beta_1$ from 0.1 to 0.9 has a greater impact on core enterprises. When $\beta_1 = 0.1$, the moment when the proportion of core enterprises using the non-repayment strategy first reaches zero is $t = 0.63$, and the moment when the proportion of MSMEs using the default strategy first reaches zero is $t = 0.21$. When $\beta_1 = 0.9$, the moment when the proportion of core enterprises using the non-repayment strategy first reaches zero is $t = 0.17$, a reduction of 0.46, and the moment when the proportion of MSMEs using the default strategy first reaches zero is $t = 0.27$, an increase of 0.46. The moment when the proportion first reaches zero is $t = 0.27$, an increase of 0.06. Thus, there is a contradiction between the change in the proportion of core enterprises incentivized by trustworthiness and the rate at which the core and MSME strategies converge to zero. Increasing the blockchain incentive core enterprise share $\beta_2$ from 0.1 to 0.9 has a greater impact on core enterprises and a smaller impact on MSMEs. When $\beta_2 = 0.1$, the moment when the proportion of core enterprises adopting a non-repayment strategy first reaches zero is $t = 0.44$, and the moment when the proportion of MSMEs adopting a default strategy first reaches zero is $t = 0.29$. When $\beta_2 = 0.9$, the moment when the proportion of core enterprises adopting a non-repayment strategy first reaches zero is $t = 0.22$, a reduction of 0.22, and the moment when the proportion of MSMEs adopting a default strategy first reaches zero is $t = 0.19$, a shortening of 0.1.

In terms of the degree of change in strategy, for core enterprises, the proportion choosing not to repay reaches a maximum of 97.21% at moment $t = 0.18$ when $\beta_1 = 0.1$. When $\beta_1 = 0.9$, this proportion drops to 2.58%, a decrease of 94.63%. For MSMEs, the proportion choosing to default reaches a maximum of 50.37% at moment $t = 0.02$ when $\beta_1 = 0.1$, and this proportion falls to 34.65% when $\beta_1 = 0.9$, a decrease of 15.72%. For core enterprises, the proportion choosing not to repay reaches a maximum of 69.28% at moment $t = 0.03$ when $\beta_2 = 0.1$. When $\beta_2 = 0.9$, this proportion drops to 32.48%, a decrease of 36.8%. For MSMEs, the proportion choosing to default reaches a maximum of 54.88% at moment $t = 0.02$ when $\beta_2 = 0.1$, and this proportion drops to 31.28% when $\beta_2 = 0.9$, a decrease of 23.6%.

When the proportion of core enterprises with creditworthiness incentives $\beta_1$ increases, core enterprises are more inclined to repay the loan, and MSMEs are almost unaffected by the change in the proportion of creditworthiness incentives. Therefore, when financial institutions regulate the creditworthiness incentive, they should give priority to regulating the impact of $\beta_1$ on core enterprises.

As shown in Tables 2 and 3, when financial institutions are committed to increasing the rate of convergence of subject strategies to reach a fully cooperative state more quickly, financial institutions are more sensitive to the information-sharing risk factor, and core enterprises are more sensitive to the share of creditworthiness incentives. When financial institutions are more concerned about the degree of change in subject strategies, financial institutions are highly sensitive to the information-sharing risk factor, and core enterprises are highly sensitive to the share of trustworthy incentives. Therefore, improving the security of blockchain information sharing and increasing the share of trustworthy incentives for core enterprises can significantly reduce the time to reach a steady state.

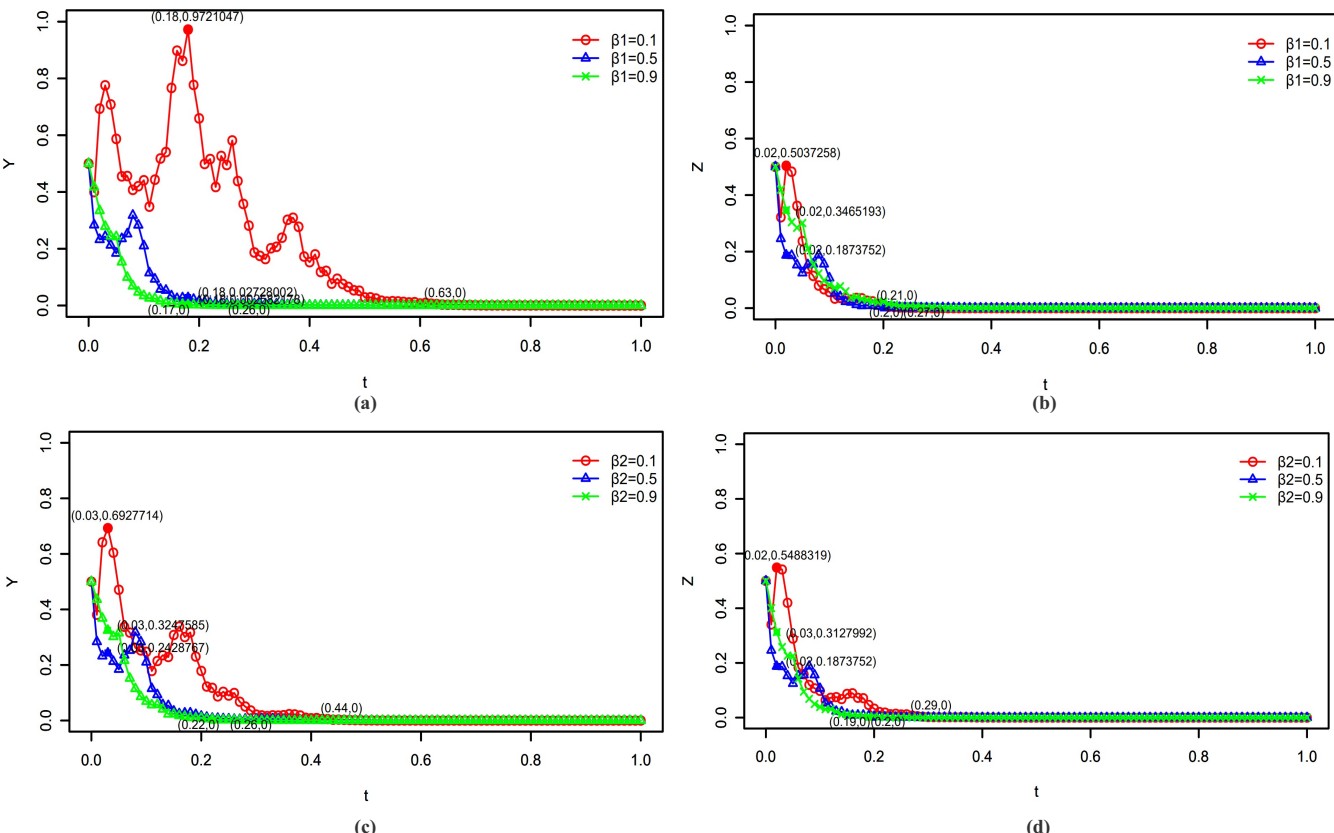

**Figure 5.** The impact of incentive share on the gaming strategies of core enterprises and MSMEs. (**a**) The impact of credit-keeping incentive share $\beta_1$ on the gaming strategies of core enterprises. (**b**) The impact of credit-keeping incentive share $\beta_1$ on the gaming strategies of MSMEs. (**c**) The impact of blockchain incentive share $\beta_2$ on the gaming strategies of core enterprises. (**d**) The impact of blockchain incentive share $\beta_2$ on the gaming strategies of MSMEs.

**Table 2.** Influence of parameters on the speed of convergence rate of subject strategy.

| Value of Parameters | The Moment of the Proportion of Noncooperative Strategies First Reaching Zero | Change in Convergence Rate |
|---|---|---|
| $\delta : 0.1 \rightarrow 0.5 \rightarrow 0.9$ | $t_x : 0.21 \rightarrow 0.36 \rightarrow 0.53$<br>$t_y : 0.42 \rightarrow 0.26 \rightarrow 0.23$<br>$t_z : 0.29 \rightarrow 0.20 \rightarrow 0.18$ | $\Delta t_x = 0.32 \uparrow$<br>$\Delta t_y = 0.19 \downarrow$<br>$\Delta t_z = 0.11 \downarrow$ |
| $\alpha : 0.1 \rightarrow 0.5 \rightarrow 0.9$ | $t_x : 0.30 \rightarrow 0.36 \rightarrow 0.32$<br>$t_y : 0.42 \rightarrow 0.26 \rightarrow 0.23$<br>$t_z : 0.29 \rightarrow 0.20 \rightarrow 0.18$ | $\Delta t_x = 0.02 \uparrow$<br>$\Delta t_y = 0.19 \downarrow$<br>$\Delta t_z = 0.11 \downarrow$ |
| $\gamma : 0.1 \rightarrow 0.5 \rightarrow 0.9$ | $t_x : 0.23 \rightarrow 0.36 \rightarrow 0.43$<br>$t_y : 0.42 \rightarrow 0.26 \rightarrow 0.23$<br>$t_z : 0.29 \rightarrow 0.20 \rightarrow 0.18$ | $\Delta t_x = 0.20 \uparrow$<br>$\Delta t_y = 0.19 \downarrow$<br>$\Delta t_z = 0.11 \downarrow$ |
| $\beta_1 : 0.1 \rightarrow 0.5 \rightarrow 0.9$ | $t_y : 0.63 \rightarrow 0.26 \rightarrow 0.17$<br>$t_z : 0.21 \rightarrow 0.20 \rightarrow 0.17$ | $\Delta t_y = 0.46 \downarrow$<br>$\Delta t_z = 0.04 \downarrow$ |
| $\beta_2 : 0.1 \rightarrow 0.5 \rightarrow 0.9$ | $t_y : 0.42 \rightarrow 0.26 \rightarrow 0.22$<br>$t_z : 0.29 \rightarrow 0.20 \rightarrow 0.19$ | $\Delta t_y = 0.20 \downarrow$<br>$\Delta t_z = 0.10 \downarrow$ |

**Table 3.** Impact of incentive and penalty parameters on the changing degree of subject strategy.

| Value of Parameters | Maximum Proportion of Noncooperative Strategies | Change in Strategy Proportion |
|---|---|---|
| $\delta : 0.1 \rightarrow 0.5 \rightarrow 0.9$ | $x(0.03) : 29.91\% \rightarrow 25.57\% \rightarrow 79.61\%$<br>$y(0.03) : 68.71\% \rightarrow 24.29\% \rightarrow 32.89\%$<br>$z(0.02) : 55.21\% \rightarrow 18.74\% \rightarrow 31.07\%$ | $\Delta x = 49.70\%\uparrow$<br>$\Delta y = 38.82\%\downarrow$<br>$\Delta z = 24.14\%\downarrow$ |
| $\alpha : 0.1 \rightarrow 0.5 \rightarrow 0.9$ | $x(0.03) : 39.26\% \rightarrow 25.57\% \rightarrow 63.62\%$<br>$y(0.03) : 68.85\% \rightarrow 24.29\% \rightarrow 32.82\%$<br>$z(0.02) : 55.23\% \rightarrow 18.74\% \rightarrow 31.03\%$ | $\Delta x = 24.36\%\uparrow$<br>$\Delta y = 36.03\%\downarrow$<br>$\Delta z = 24.20\%\downarrow$ |
| $\gamma : 0.1 \rightarrow 0.5 \rightarrow 0.9$ | $x(0.03) : 34.84\% \rightarrow 25.57\% \rightarrow 70.54\%$<br>$y(0.03) : 68.79\% \rightarrow 24.29\% \rightarrow 32.89\%$<br>$z(0.02) : 55.22\% \rightarrow 18.74\% \rightarrow 31.05\%$ | $\Delta x = 35.70\%\uparrow$<br>$\Delta y = 38.85\%\downarrow$<br>$\Delta z = 24.17\%\downarrow$ |
| $\beta_1 : 0.1 \rightarrow 0.5 \rightarrow 0.9$ | $y(0.18) : 97.21\% \rightarrow 2.73\% \rightarrow 2.58\%$<br>$z(0.02) : 50.37\% \rightarrow 18.74\% \rightarrow 34.65\%$ | $\Delta y = 94.63\%\downarrow$<br>$\Delta z = 15.72\%\downarrow$ |
| $\beta_2 : 0.1 \rightarrow 0.5 \rightarrow 0.9$ | $y(0.03) : 69.28\% \rightarrow 24.29\% \rightarrow 32.48\%$<br>$z(0.02) : 54.88\% \rightarrow 18.74\% \rightarrow 31.28\%$ | $\Delta y = 36.80\%\downarrow$<br>$\Delta z = 23.60\%\downarrow$ |

## 6. Conclusions

Based on the traditional three-party evolutionary game model for receivables financing in supply chain finance, we transform blockchain features into parameters to join the mathematical model, analyze the stability of strategy choice of game subjects under stochastic situations by constructing a stochastic evolutionary game model among financial institutions, core enterprises and MSMEs, and further quantitatively analyze the influence of regulatory strength, credit sharing risk coefficient, credit sharing incentive coefficient, and incentive share on subjects' strategy choice. We have drawn the following conclusions from our study of the impact of blockchain technology on the receivables financing decisions of financial institutions, core enterprises, and MSMEs: Firstly, blockchain technology ensures open and transparent information for all parties involved in receivables financing, and information sharing will bring certain risks and incentives to each participating entity. When the risk and incentive of information sharing brought by the interfacing blockchain platform are high, the likelihood of core enterprises and MSMEs keeping their trust increases because as the information sharing among participating parties becomes deeper, the probability of being punished for non-compliance increases, and the positive incentive brought by trustworthiness motivates enterprises to continue to choose trustworthiness in the next cooperation. However, for financial institutions, continuing to interface with blockchain platforms is difficult to achieve when the risk of information sharing is high, even if information sharing provides them with some incentive.

Secondly, when not connected to a blockchain platform, financial institutions pay extra regulatory costs to audit the authenticity of transactions and receivables. When financial institutions are more heavily regulated, core and MSMEs are more likely to be trustworthy, as their non-compliance is then more likely to be detected and thus penalized. At the same time, financial institutions tend not to interface with blockchain platforms because the gains from blockchain technology are smaller under sufficiently strong regulation, and the costs required to build a blockchain platform are higher. Thirdly, for core enterprises and MSMEs, the act of keeping trust will bring incentives, and they will obtain extra incentives when they choose to keep trust after connecting to the blockchain platform. When a core enterprise or MSME chooses to keep its trust during the cooperation process, it will be trusted by financial institutions, which grant it certain preferential policies, such as lower interest rates on loans. When the incentive to keep trust increases, the likelihood of core enterprises keeping trust increases significantly, while there is little chance for MSMEs. As the originator of receivables financing and the ultimate repayer, when MSMEs default, the core enterprises still have to repay the loan without downstream capital inflows, and even if they default, they cannot earn investment returns. In contrast, when the core enterprise defaults, the MSME still receives a grant from the financial institution. Therefore, the benefits from the credit compliance incentive are more important to the core enterprises.

Financial institutions can formulate more favorable policies to increase the credit-keeping incentive and strengthen the cooperation with core enterprises to promote the long-term operation of supply chain finance. In this paper, the following aspects will be further investigated in depth in the course of subsequent research. Firstly, we have used numerical simulations to study the three-party game, which will be more convincing if we collect enough data to support them with real cases. Secondly, the operation process of supply chain finance is disturbed by many complex random factors, and the explanation of random factors in this paper is not comprehensive enough. Finally, we have limited the financing process of supply chain finance to the pledged receivables financing model and only considered three participants, namely financial institutions, core enterprises, and MSMEs, and more complex scenarios can be considered in the next study.

**Author Contributions:** Conceptualization, Q.Z.; methodology, Q.Z.; software, R.Z.; validation, M.X.; formal analysis, R.Z.; writing—original draft preparation, R.Z.; writing—review and editing, Q.Z.; supervision, M.X. All authors have read and agreed to the published version of the manuscript.

**Funding:** This research is supported by the National Natural Science Foundation of China (11671229; 11971259); Natural Science Foundation of Shandong Province of China (ZR2022MA029; ZR2020MA032); Graduate Education Quality Improvement Program of Shandong Province (SDYKC19197); and Key Research Project of Financial Application in Shandong Province (2019-JRZZ-15).

**Institutional Review Board Statement:** Not applicable.

**Informed Consent Statement:** Not applicable.

**Data Availability Statement:** Not applicable.

**Conflicts of Interest:** The authors declare no conflict of interest.

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
