# Peer review of "Three-Party Stochastic Evolutionary Game Analysis of Supply Chain Finance Based on Blockchain Technology"

_sustainability, doi:10.3390/su15043084_

Round 1
Reviewer 1 Report
This is a very nice paper. I would suggest the authors expand their analysis to a more general setup with detailed discussion.
Author Response
Point: This is a very nice paper. I would suggest the authors expand their analysis to a more general setup with detailed discussion.
Response: Thanks very much for taking your time to review this manuscript. We really appreciate all your comments and suggestions. We only consider the impact of blockchain technology on the evolutionary game of supply chain finance players under the accounts receivable financing model with the introduction of stochastic factors. In future research, we plan to consider a more comprehensive set of factors to more fully understand the impact of blockchain technology on supply chain finance.

Reviewer 2 Report
This study develops a three-party stochastic evolutionary game mode to discuss the integration of blockchain technology with a supply chain finance platform around the fraudulent default phenomenon in supply chain finance receivables financing. The paper is well organized, and can be published after following issues are resolved.
(1)Research motivitions could be highlighted in the first section.
(2)There lacks of literature review section, since there are many blockchain-enabled applications and supply chain management literatures.
(3)In section 2.3, there may be an error in the second paragraph between the core enterprise's and MSME's return.
(4)When the financial institution adopts the strategy of "docking", why do core enterprises and MSME need to pay the cost of using the blockchain?
(5)What is the links of Section 2 and Section 3? Why there are two model sections?
(6) It will be better if the practical implications could be addressed in the discussion section.
(7)There is no literature review section. There are vast majority of related publications need to be reviewed, such as:
1) Blockchain-enabled cross-border e-commerce supply chain management: A bibliometric systematic review.
2)Blockchain technology for supply chain management: A comprehensive review.
3)Blockchain technology for secure supply chain management: A comprehensive review.
...
Reviewer 3 Report
The authors have presented a very interesting contribution to supply chain finance. I have a minor comment: as a reader, I would expect a few lines in the introduction about blockchain technology and its application. The authors could refer to any recent relevant reviews for this purpose.
Author Response
Point: The authors have presented a very interesting contribution to supply chain finance. I have a minor comment: as a reader, I would expect a few lines in the introduction about blockchain technology and its application. The authors could refer to any recent relevant reviews for this purpose.
Response: Thanks very much for taking your time to review this manuscript. Indeed, as the reviewer mentioned, we have reorganized the structure and content of the article by adding an example of supply chain finance fraud and two examples of blockchain technology enabling supply chain finance in Section 1, as follows:
1) From September to November 2016, Genertec Dalian Machine Tool falsified its transaction information with BYD by falsifying accounts receivable, forging contracts and official seals, and so on, to obtain RMB 600 million from Zhongjiang Trust.
2) ICBC has built the ICBC e-Credit network financing and financial services platform to prevent performance risks through automatic execution of smart contracts and build a new and reliable supply chain credit system.
3)Tencent Cloud integrated blockchain technology with warehouse receipt pledge financing scenarios, solving the problems of forgery, multiple pledges and false collateral in the traditional warehouse receipt pledge financing process, and realizing data-based management of the pledge transaction process.
